# Design and Evaluation of a Structural Analysis-Based Fault Detection and Identification Scheme for a Hydraulic Torque Converter

**DOI:** 10.3390/s18124103

**Published:** 2018-11-23

**Authors:** Qi Chen, Jincheng Wang, Qadeer Ahmed

**Affiliations:** 1School of Mechanical Engineering, Hefei University of Technology, Hefei 230009, China; hfut_wjc@126.com; 2Center for Automotive Research, The Ohio State University, Columbus, OH 43212, USA

**Keywords:** fault detection and identification, structural analysis, hydraulic torque converter, MSO sets, residual

## Abstract

A hydraulic torque converter (HTC) is a key component in an automatic transmission. To monitor its operating status and to detect and locate faults, and considering the high-efficiency fault detection and identification (FDI) scheme design by the methodology of structural analysis (SA), this paper presents an SA-based FDI system design and validation for the HTC. By the technique of fault mode and effect analysis (FMEA), eight critical faults are obtained, and then two fault variables are chosen to delegate them. Fault detectability and isolability, coupled with different sensor placements, are analyzed, and as a result, two speed sensors and two torque sensors of pump and turbine are selected to realize the maximal fault detectability and fault isolability: all six faults are detectable, four faults are uniquely isolable, and two faults are isolated from the other faults, but not from each other. Then five minimal structurally overdetermined (MSO) sets are easily acquired by SA to generate five corresponding residuals. The proposed FDI scheme of the HTC by SA is first validated by a theoretical model, then by an offline experiment in a commercial SUV, and the testing results indicate a consistent conclusion with the simulations and theory analysis.

## 1. Introduction

A hydraulic torque converter (HTC) is used in automatic transmission (AT) to transform engine power to the transmission shafts, and then to the vehicle. Its main function is to adjust the torque and speed between the engine and transmission, as well as to avoid engine overload. So, if there is a malfunction in the HTC, it will directly affect the output torque from the engine, decrease the transferring efficiency of the transmission, and even result in gear shifting failure or power loss of the transmission, or a broken engine when it is stuck or damaged, which may cause a fatal accident. Therefore, it is urgent to produce fault detection and identification (FDI) of system and sensor faults for the HTC that meet the demands of producing ISO 26262-compliant automobiles [1].

In recent years, research with respect to the HTC has mostly focused on modelling and optimization [2,3], the strategy of the HTC clutch slip controller [4,5], factors influencing performance [6,7], transient characteristic testing [8,9], etc. It is worth mentioning that in Ref. [8], the transient performance of three torque converters from a Ford Taurus, a Honda CRV, and a Mercedes-Benz were tested to validate the proposed computer model, in which the large amount of data about sensor measurements, such as pump speed, turbine speed, pump torque, and turbine torque, also provides a reference for the experimental validation in this paper.

However, when it comes to the fault diagnosis of the HTC, related articles and studies do not provide much information. Among them, most of the reports are related to the failure mode, causes, and effects, such as in [10], in which three fault effects aroused by HTC malfunction were analyzed, and five possible fault causes resulting in lock-up clutch failure in the HTC were represented in detail. The failure mode and the causes of one-way clutch and lock-up clutch in the HTC were also discussed in [11]. In [12], four common failure modes (overheating of the HTC, oil leakage, a large vibration of lock-up connection, and abnormal noise) and their possible causes were presented, and five fault effects (decrease/loss of power, no gear shifting for driving, acceleration weakness at low speed, acceleration weakness at high speed, and engine stalling on gear) were also discussed. In [13], HTC failures were divided into three categories—fixed support system (bearings), powertrain transfer system (pump and turbine), and recycled oil circuit system (sealing)—and the causes leading to these three kinds of failure were studied. In [14], the failure mode and failure effects of the HTC and their causes were analyzed by the techniques of failure mode and effect analysis (FMEA), and fault tree analysis (FTA), where the hazard degree of different failures was evaluated, and the logic relationship between system failures, subsystem failures, and component failures was established. 

Several studies have been concerned with inspection, repair, and maintenance of the HTC. For example, in Ref. [10], six elements for inspection—the exterior, one-way clutch in the guide wheel, pump axis, internal motion interference, and clutch friction material—were discussed, to judge if there was HTC failure. In Ref. [15], besides the inspection of the exterior and the one-way clutch in the guide wheel, the amount of the bushing yaw was also examined by a dial indicator to correctly install the HTC in the engine flywheel. In Ref. [12,16], how to check and maintain the parts of the lock-up clutch and the one-way clutch in the guide wheel was discussed.

Additionally, one paper [17] focused on a real-time monitoring controlling system for HTC testing; five sensors were used to measure the temperature, oil pressure, oil flow, axis torque, and speed in the HTC. If the temperature was too high or of the load was too much, a protection system was initiated to shut off the power for safety.

Based on the above investigation of HTC fault diagnosis, we know that existing reports are more concerned with analyzing failure mode and causes, aiming to help technicians to easily find faults in the HTC, and then to rapidly repair them. The techniques in these papers are mostly based on experience or professional instruments, and they are geared toward practical application, but theoretical research is rare. In addition, most inspections need to remove the HTC from the vehicle, and offline fault detection is required.

To resolve the above issues, this paper is committed to theoretical research on fault diagnosis for the HTC, so as to monitor the running state in real time, identify the faults immediately, and to help to realize a reliable controlling system with the function of fault tolerance. We consider that the model-based methodology [18,19] is effective in fault detection and identification (FDI) for automotive systems, such as the suspension system [20], the hydraulic braking system [21], the steer-by-wire system [22,23], the electrical steering system [24], etc. Among these reports, some specific techniques based on models from the system physical structure, such as parameter estimation, parity equation, observer, cumulative sum (CUSUM), support vector machines (SVM), or probabilistic neural networks (PNN), were employed for fault detection and fault isolation. Of course, we can also use these techniques for fault diagnosis of the HTC, but after investigating these papers, we found little use for a systematic fault detectability (FD) and fault isolability (FI) analysis. In addition, the procedures for performing the FDI scheme of generating residuals are not concise. However, we found that another model-based theory, structural analysis (SA), has the virtue of intuitively making a detectability and isolability analysis, and realizing an efficient FDI scheme design for a complex system. The theory of SA is based on the model structure represented by a bipartite graph, which is possible thanks to the conception of a bond graph [25,26]. It was previously successfully applied in fault diagnosis and FDI system design for complicated systems, including linear and nonlinear systems, in Ref. [27,28,29,30], and was also used in vehicle powertrain systems [31,32,33,34]. Based on those successful applications of SA, we also use this method to perform an efficient FDI scheme for the HTC in this paper.

This paper presents a systematic approach to performing efficient FDI system design for the HTC based on the theory of SA, where first the techniques of structure representation (SR) and Dulmage–Mendelsohn (DM) composition will be used to intuitively analyze the FD and FI, then an efficient sensor placement is executed to realize the optimal capability of FD and FI for the HTC, and then minimal structurally overdetermined (MSO) sets are directly obtained to generate sequential residuals, and finally an analytic redundant relationship (ARR) and observer are employed for robust residual design. The numerical simulation of an HTC model in MATLAB, as well as an experimental study on a commercial vehicle, show that the proposed FDI system can detect and isolate the desired faults, which is consistent with the theoretical result by SA. In this paper, we show a general interpretation of applying SA in the case of HTC fault diagnosis, which also establishes a reference for the fault diagnosis of other mechatronics systems using SA in the future.

The rest of this paper is organized as follows: Hazard analysis of the HTC based on FMEA is described in Section 2, where eight critical faults are obtained. Fault diagnosis based on structural analysis (SA) is performed in Section 3, which presents the fault modelling of the HTC, and key procedures like DM decomposition, fault detectability (FD) analysis, fault isolability (FI) analysis, MSO sets, and residual design. An FDI system is designed and simulated in Section 4 to verify the correctness of the SA methodology. Experimental validation is presented in Section 5, and a summary is provided in Section 6.

## 2. Hazard Analysis of the Hydraulic Torque Converter

### 2.1. Basic Structure and Function of the HTC

The hydraulic torque converter is a nonrigid transmission part with automatic transmission fluid (ATF) oil as the working medium. It is usually located at the front of the automatic transmission and is connected to the engine flywheel. Figure 1 gives a sketch of a typical HTC structure, which is primarily made up of the pump (impeller), turbine, stator (guide wheel) with one-way clutch, lock-up clutch, etc.

### 2.2. Fault Mode and Effect Analysis (FMEA) of the HTC 

In order to obtain comprehensive results of the hazard analysis of the HTC, we employ the technique of a fault mode and effect analysis (FMEA) to display all the possible faults. FMEA is a logical induction that uses a bottom-up strategy to explore potential fault patterns, analyze the system within the scope of the various components of the potential fault, study the impact and causes of the fault, and evaluate the level of risk of danger [35,36]. Through analyzing the function relation among the elements, we can identify the possibility of propagation of each type of failure and predict its effects on system performance, which can help us to obtain critical faults in the system.

By employing the FMEA, the severity (S) of each fault, the occurrence (O) frequency of the fault, and the detection (D) of the fault are evaluated and represented by numbers (1 to 10), and then a risk priority number (RPN) by S × O × D is used to determine the risk level of each fault. The higher the RPN, the more severe the fault. Table 1 presents the FMEA analysis results for the HTC [10,11,12,14].

From Table 1, we can see the top eight faults according to RPN value, so these are defined as critical faults in the HTC. According the location of the fault, we can divide them into two groups, the pump group and the turbine group. The faults in the pump group are pump blade fracture, damaged seals/oil leakage, disconnection between the pump wheel and flywheel, and stuck lock-up clutch separation. The faults in the turbine group are turbine blade fracture, spline broken in the turbine wheel, damaged guild ring in the stator, and lock-up clutch connection failure.

When these eight faults happen, it may cause full loss of the HTC, which may result in the malfunction of the automatic transmission and the vehicle, and even fatal accidents. Thus, we should work out a fault diagnosis system to detect them; to identify which fault happened, we also need to isolate them. In the next section, we will discuss a model-based method for detecting and identifying faults in an HTC based on SA technique.

## 3. Fault Diagnosis of the Hydraulic Torque Converter via Structural Analysis

Structural analysis (SA) [37] is an effective model-based diagnosis method for fault detection and identification. The approach utilizes graphic tools to efficiently evaluate fault detectability and isolability of the system, and it constructs the minimum set of overdetermined (MSO) equations conveniently, for consequential residual design. Based on our previous successful experience in automatic manual transmission (AMT) [34,38] by employing SA, here, we present another application of SA in the HTC.

Figure 2 presents the main steps for SA execution. We will demonstrate the procedures thoroughly when applying SA in the fault diagnosis of the HTC, in the following sections.

### 3.1. Fault Modeling of the HTC

#### 3.1.1. Mathematical Model of the HTC

Fault modelling can be generated by combining the system model with some fault variables that can delegate the critical faults. Here, we first study the system modelling of the HTC. Figure 3 shows a typical diagram of an HTC that is used in a vehicle, where we can see that the HTC is transferring speed and torque from the engine to the transmission, then to the wheels for vehicle driving. Based on the literature [8,39], we can establish the model of the torque converter by Equations (1)–(5):(1)$$\;T_{tp}={\left(\frac{\omega_{p}}{KF\left(SR\right)}\right)}^{2}\;$$
(2)$$\;T_{tt}=TR\left(SR\right)\cdot T_{tp}\;$$
(3)$$\;T_{p}=T_{tp}+{\dot{\omega }}_{p}\cdot I_{p}\;$$
(4)$$\;T_{t}=T_{tt}-{\dot{\omega }}_{t}\cdot I_{t}\;$$
(5)$$\;SR=\omega_{t}/\omega_{p}\;$$
where KF is the K-factor, TR is the torque ratio, and SR is the speed ratio.

Note that there are two ways to describe the relationship between $T_{tp}$ and $\omega_{p}$. One is by using the K-factor, in Equation (1); the other is by using the capacity factor (CF); the equation is $T_{tp}=CF\left(SR\right)\cdot \omega_{p}^{2}$. Here, we use the K-factor because we have the experimental data of the K-factor from later testing.

#### 3.1.2. Fault Modelling of the HTC

According to FMEA, we know there are eight critical faults in the HTC. Here, we introduce two fault variables, $f_{KF}$ and $f_{TR}$, to represent the critical faults lying in the pump group and turbine group, respectively. Table 2 summarizes the critical faults and the related denoted variables.

Based on the faults in Table 2 and Equations (1)–(5), we can obtain the fault modelling shown in Equation (6):(6)$$\;\{\begin{array}{l} e_{1}:T_{tp}={\left(\frac{\omega_{p}}{KF\left(SR\right)\cdot f_{KF}}\right)}^{2}\\ e_{2}:T_{tt}=TR\left(SR\right)\cdot T_{tp}\cdot f_{TR}\\ e_{3}:T_{p}=T_{tp}+{\dot{\omega }}_{p}\cdot I_{p}\\ e_{4}:T_{t}=T_{tt}-{\dot{\omega }}_{t}\cdot I_{t}\\ e_{5}:SR=\frac{\omega_{t}}{\omega_{p}}\\ e_{6}:y_{\omega }{}_{p}=\omega_{p}+f_{\omega }{}_{p} \end{array}\;\;\;\;\;$$

Here, we see that the two faults ($f_{KF}$ and $f_{TR}$) are gain type, because they represent the fault state of the related faults in Table 2.

First, in terms of $f_{KF}$, the range of which is 1 to +∞, when it is equal to 1, there is no fault. With an increase of $f_{KF}$, the value of $T_{tp}$ will decrease, so that a fault may occur, such as pump blade fracture, oil leakage, or loosening of the connection between the pump wheel and fly wheel. When it tends to +∞, which will result in $T_{tp}$ being zero, stuck lock-up clutch separation appears.

Second, with regard to $f_{TR}$, we set the range as 0 to 1. When it is 1, it is healthy, but when it decreases, the value of $T_{tt}$ will drop, so that a fault may occur, such as turbine blade fracture, broken spline, or guild ring damage; when it is 0, a complete fault happens, such as a missing lock-up clutch connection.

### 3.2. Structural Representation of the HTC

Structural representation is used to describe the correspondence between all of the variables and the equations contained in the system [37]. Figure 4 shows a structural representation of the HTC fault model by Equation (6). Here, we divide the variables in the model into three groups:
Unknown variables: $\left\{T_{tp},T_{tt},T_{p},T_{t},\omega_{p},\omega_{t},SR\right\}$Known variables: $\left\{y_{\omega_{p}}\right\}$Fault variables: $\left\{f_{KF},f_{TR},f_{\omega }{}_{p}\right\}$

### 3.3. Analysis of Fault Detectability and the Isolability of the HTC

#### 3.3.1. Fault Detectability Analysis

Fault detectability (FD) means that a fault is detectable when it occurs in the system. According to reference [28], Dulmage–Mendelsohn (DM) decomposition [40] is a mathematical tool that is used to rearrange the equations by the fault model into the form of a bipartite graph. After doing a DM decomposition for a fault model, we divided the equations into three groups: the underdetermined part ($M^{-}$), the just-determined part ($M^{0}$), and the overdetermined part ($M^{+}$), shown in Figure 5. Here, the underdetermined part $M^{-}$, the just-determined part $M^{0}$, and the overdetermined part $M^{+}$ means that the number of equations is less than, equal to, and more than the unknown variables in the equations, respectively. If a fault lies in the underdetermined part ($M^{-}$) or in the just-determined part ($M^{0}$), it is not detectable, because there is no more equation as a redundancy; if it lies in the overdetermined part ($M^{+}$), it is detectable, because we have redundant equations there.

After applying DM decomposition on the fault modelling at Equation (6), we obtain the DM decomposition of the HTC in Figure 6a. The result shows that there is only one part here, and that it is the $M^{-}$ part, so that the number of equations is less than that of the variables, and all of the faults are not detectable.

#### 3.3.2. Fault Isolability Analysis

Fault isolability (FI) means that when the fault occurs, it can be isolated and located among other faults. Based on the strategy in [28] and the same methodology in our previous study in AMT [38], we can obtain a fault isolability matrix (FIM); in an FIM, if a fault only exists with self-correlation, the fault can be isolated, and if a fault and other faults exist with cross-correlation, the fault cannot be isolated.

Figure 6b gives the results of a FIM of the HTC, where we can see that all of the faults are correlated with other faults, so that they are not dependent and are not isolable.

### 3.4. Sensor Placement for the HTC

An effective way to improve fault detectability and isolability is to add sensors to the HTC system. Based on the fault model in Equation (6) and considering that the two sensors (torque transferred by the pump ($T_{tp}$) and torque transferred by the turbine ($T_{tt}$)) are impractically installed, we obtain three unknown variables where the sensors can be placed. They are $\left\{\omega_{t},T_{p},T_{t}\right\}$, so that all together there are $C_{3}^{1}+C_{3}^{2}+C_{3}^{3}=7$ combinations when choosing one to three sensors. It is hard to calculate the FD and FI analysis one-by-one via SA, so we developed a sensor placement tool to implement this job [29,38].

When a sensor is placed, a sensor fault is involved because the sensor measurement can also have a fault, so with an increased number of sensors, the fault frequency also rises. Thus, it is not true that more sensors are better. We need to evaluate the FD and FI with different sensor sets by synthetically considering their capability, as well as the number of sensors. The best scheme is to use the minimum amount of sensors to reach maximal FD and FI.

Table 3 shows the results of FD and FI after conducting sensor placement. We can clearly see that group #4, #6 and #7 can detect all of the faults, but that group #7 is optimal, where not only are all faults detectable, but they are the most isolable faults.

Thus, the conclusion of the sensor placement for the HTC is that placing four sensors may make all six faults detectable, and that five fault sets are isolable, whereas four faults are uniquely isolable, leaving two faults ($f_{T_{t}},f_{TR}$) that are isolable from the other faults, but not from each other.

Sensor No., number of the sensor; FT.No., number of the fault; Det.FT.No., number of the detectable fault; UnDeT.FT.No., number of the undetectable fault; Iso.FT.Set.No., number of the isolable fault set; Uni.Iso.FTs.No., number of the uniquely isolable fault; Undet.Fault.List, list of the undetectable fault; Iso.Fault.Sets List, list of the isolable fault sets; Unique Iso.Fault.List, list of the uniquely isolable faults.

Equation (7) gives the updated fault model of the HTC with four sensors installed, in which there are six faults: the two system faults ($f_{KF},f_{TR}$) from the HTC system, and four sensor faults ($f_{\omega_{p}\;},f_{\omega_{t}\;},f_{T_{p}\;},f_{T}{}_{t}$) from the possible malfunction of sensor measurements:
(7)$$\left\{ \begin{array}{l} \;e_{1}:T_{tp}={\left(\frac{\omega_{p}}{KF\left(SR\right)\cdot f_{KF}}\right)}^{2}\;\\ \;e_{2}:\;T_{tt}=TR\left(SR\right)\cdot T_{tp}\cdot f_{TR}\;\\ \;e_{3}:T_{p}=T_{tp}+{\dot{\omega }}_{p}\cdot I_{p}\;\\ \;e_{4}:T_{t}=T_{tt}-{\dot{\omega }}_{t}\cdot I_{t}\;\\ \;e_{5}:SR=\frac{\omega_{t}}{\omega_{p}}\\ \;e_{6:}\;y_{\omega }{}_{p}=\omega_{p}+f_{\omega }{}_{p}\;\\ \;e_{7:}\;y_{\omega }{}_{t}=\omega_{t}+f_{\omega }{}_{t}\;\\ \;e_{8:}\;y_{T}{}_{p}=T_{p}\cdot f_{T}{}_{p}\;\\ \;e_{9:}\;y_{T}{}_{t}=T_{t}\cdot f_{T}{}_{t}\; \end{array}\right.$$
where $y_{\omega_{p}},\;y_{\omega_{t}},\;y_{T_{p}}$, $y_{T_{t}}$ are the measurements of the pump angular velocity, turbine angular velocity, pump torque, and turbine torque, respectively; and $f_{\omega_{p}\;},f_{\omega_{t}\;},f_{T_{p}\;},f_{T}{}_{t}$, are the fault variables corresponding to $\omega_{p}$,$\;\omega_{t},T_{p}$, and $T_{t}$. Here, we assume that the velocity faults ($f_{\omega_{p}\;},f_{\omega_{t}\;}$) are bias-type, and that the torque faults ($f_{T_{p}\;},f_{T}{}_{t}$) are gain-type, based on the possible faults in sensors, such as no signal, deviation, or drift in the measurement.

Note: Here, the fault type of the speed sensor faults ($f_{\omega_{p}\;},f_{\omega_{t}\;}$) may be gain-type, and the torque sensor faults ($f_{T_{p}\;},f_{T}{}_{t}$) may be bias-type. We show just one of the possible cases for demonstration.

Figure 7 intuitively shows the result of FD and FIM related to the fault model by Equation (7), where we can see intuitively from Figure 7a that all of the faults are detectable, because they all lie in the overdetermined part ($M^{+}$) where the equations are more than the variables; in addition, we can also clearly observe from Figure 7b that five sensors sets are isolated, in which four faults ($f_{KF},f_{\omega_{p}\;},f_{\omega_{t}\;},f_{T_{p}\;}$) are uniquely isolable, and two sensors ($f_{KF},f_{TR}$) are isolable from the other faults but not distinguished from each other.

### 3.5. Finding MSO Sets

According to the research in [41], a minimal structurally overdetermined (MSO) set is a collection of minimal equations to generate one residual. The number of MSO sets is the amount of residuals, which are the necessary and minimal size of residuals to reach the optimal ability of fault detection and isolation in a system. Through the algorithms in [41,42], we can obtain all five MSO sets for the HTC, corresponding to the model of Equation (7). Table 4 shows the MSO sets of the HTC, where we can also obtain the detection information of the faults related to every MSO set.

In the table, the symbol “●” indicates that the fault is detectable, and “×” means the fault is not detectable. For example, $T_{1}$ can detect five faults ($f_{\omega_{p}\;},f_{\omega_{t}\;},f_{T_{p}\;},f_{T}{}_{t},f_{TR}$), but it cannot detect ($f_{KF}$).

### 3.6. Residual Design

According to [37], a testable set of equations can generate a residual. According to Table 4, we obtain five testable sets of equations ($T_{1}$ to $T_{5}$), which can produce five residuals correspondingly. In this section, we will discuss the detailed procedures of residual design.

(1) Residual-1

Residual-1 is derived from $T_{1}$, which consists of eight equations: $\left\{e_{2},e_{3},e_{4},e_{5},e_{6},e_{7},e_{8},e_{9}\right\}$.
(8)$$\;\{\begin{array}{l} e_{2}:\;T_{tt}=TR\left(SR\right)\cdot T_{tp}\\ e_{3}:T_{p}=T_{tp}+{\dot{\omega }}_{p}\cdot I_{p}\\ e_{4}:T_{t}=T_{tt}-{\dot{\omega }}_{t}\cdot I_{t}\\ e_{5}:SR=\frac{\omega_{t}}{\omega_{p}}\\ e_{6:}\;y_{\omega }{}_{p}=\omega_{p}\\ e_{7:}\;y_{\omega }{}_{t}=\omega_{t}\\ e_{8:}\;y_{T}{}_{p}=T_{p}\\ e_{9:}\;y_{T}{}_{t}=T_{t} \end{array}\;\;\;\;\;\;\;\;\;\;\;\;\;\;\;\;\;$$

According to the research by Nyberg and Frisk [43], the above equations can yield many possible residuals, but in order to avoid any derivative part in the residuals, here, we also employ the technique of an analytic redundant relationship (ARR) [44] to generate residual-1. After this, we substitute all the other equations into $e_{4}$, and we obtain an ARR, shown as:(9)$$\;I_{t}\cdot {\dot{y}}_{\omega_{t}}+TR\left(SR\right)\cdot I_{p}\cdot {\dot{y}}_{\omega }{}_{p}+y_{T}{}_{t}-TR\left(SR\right)\cdot y_{T}{}_{p}=0\;$$

This ARR can generate residual-1 in state–space form, as given by:(10)$$\;\{\begin{array}{l} \dot{x}=-\beta_{1}\left(x+I_{t}\cdot y_{\omega_{t}}+TR\left(SR\right)\cdot I_{p}\cdot y_{\omega }{}_{p}\right)+y_{T}{}_{t}-TR\left(SR\right)\cdot y_{T}{}_{p}\\ r_{1}=x+I_{t}\cdot y_{\omega_{t}}+TR\left(SR\right)\cdot I_{p}\cdot y_{\omega }{}_{p} \end{array}\;$$

Here, $\beta_{1}$ should be more than 0 for the stability of the system [37], and the same requirements hold with respect to $\beta_{2},\;\beta_{3},$
$\beta_{4}$, and $\beta_{5}$.

(2) Residual-2

Residual-2 is derived from $T_{2}$, which consists of eight equations:$\;\left\{e_{1},e_{2},e_{3},e_{4},e_{5},e_{7},e_{8},e_{9}\right\}$.
(11)$$\;\{\begin{array}{l} e_{1}:T_{tp}={\left(\frac{\omega_{p}}{KF\left(SR\right)}\right)}^{2}\\ e_{2}:\;T_{tt}=TR\left(SR\right)\cdot T_{tp}\\ e_{3}:T_{p}=T_{tp}+{\dot{\omega }}_{p}\cdot I_{p}\\ e_{4}:T_{t}=T_{tt}-{\dot{\omega }}_{t}\cdot I_{t}\\ e_{5}:SR=\frac{\omega_{t}}{\omega_{p}}\\ e_{7:}\;y_{\omega }{}_{t}=\omega_{t}\\ e_{8:}\;y_{T}{}_{p}=T_{p}\\ e_{9:}\;y_{T}{}_{t}=T_{t} \end{array}\;\;\;\;\;\;\;\;\;\;\;\;\;\;\;\;\;$$

Here, we cannot obtain an ARR directly, because we do not know the value of $\omega_{p}$, so we first substitute $e_{1}$, $e_{5}$,${\;e}_{7}$, $e_{8}$ into $e_{3}$ to get:(12)$$\;{\dot{\omega }}_{p}\cdot I_{p}=y_{T_{p}}-{\left(\frac{\omega_{p}}{KF\left(\frac{y_{\omega_{t}}}{\omega_{p}}\right)}\right)}^{2}\;$$

It is hard to obtain the value of $\omega_{p}$ in the form of a mathematical equation, but we can employ a fourth-/fifth-order Runge–Kutta numerical integration algorithm to resolve Equation (12), which is not difficult to implement in MATLAB. Here we use ${\hat{y}}_{\omega_{p}}$ as the answer of $\omega_{p}$. Then, choosing $e_{4}$ as an ARR (Equation (13)), we obtain residual-2 in state–space form, as given by Equation (14):(13)$$\;I_{t}\cdot {\dot{y}}_{\omega_{t}}+y_{T}{}_{t}-TR\left(\frac{y_{\omega_{t}}}{{\hat{y}}_{\omega_{p}}}\right)\cdot {\left(\frac{\;{\hat{y}}_{\omega_{p}}}{KF\left(\frac{y_{\omega_{t}}}{{\hat{y}}_{\omega_{p}}}\right)}\right)}^{2}=0\;$$
(14)$$\;\{\begin{array}{l} \dot{x}=-\beta_{2}\left(x+I_{t}\cdot y_{\omega_{t}}\right)+y_{T}{}_{t}-TR\left(\frac{y_{\omega_{t}}}{{\hat{y}}_{\omega_{p}}}\right)\cdot {\left(\frac{\;{\hat{y}}_{\omega_{p}}}{KF\left(\frac{y_{\omega_{t}}}{{\hat{y}}_{\omega_{p}}}\right)}\right)}^{2}\\ r_{2}=x+I_{t}\cdot y_{\omega_{t}} \end{array}\;$$

(3) Residual-3

Residual-3 is derived from $T_{3}$, which consists of eight equations: $\left\{e_{1},e_{2},e_{3},e_{4},e_{5},e_{6},e_{8},e_{9}\right\}$.
(15)$$\;\{\begin{array}{l} e_{1}:T_{tp}={\left(\frac{\omega_{p}}{KF\left(SR\right)}\right)}^{2}\\ e_{2}:\;T_{tt}=TR\left(SR\right)\cdot T_{tp}\\ e_{3}:T_{p}=T_{tp}+{\dot{\omega }}_{p}\cdot I_{p}\\ e_{4}:T_{t}=T_{tt}-{\dot{\omega }}_{t}\cdot I_{t}\\ e_{5}:SR=\frac{\omega_{t}}{\omega_{p}}\\ e_{6:}\;y_{\omega }{}_{p}=\omega_{p}\\ e_{8:}\;y_{T}{}_{p}=T_{p}\\ e_{9:}\;y_{T}{}_{t}=T_{t} \end{array}\;\;\;\;\;\;\;\;\;\;\;\;\;\;\;\;\;$$

Here we meet the same problem with residual-2, that is, we do not know $\omega_{t}$, so we adopt the same strategy by first calculating $\omega_{t}$ and then using an ARR to generate residual-3. After we substitute $e_{1}$,$e_{2}$,$e_{5}$,$e_{6},e_{9}$ to $e_{4}$, we get:(16)$$\;{\dot{\omega }}_{t}\cdot I_{t}=TR\left(\frac{\omega_{t}}{y_{\omega_{p}}}\right)\cdot {\left(\frac{y_{\omega_{p}}}{KF\left(\frac{\omega_{t}}{y_{\omega_{p}}}\right)}\right)}^{2}-y_{T}{}_{t}\;$$

By using a fourth-/fifth-order Runge–Kutta numerical integration algorithm, we can obtain the answer of $\omega_{t}$, denoted as ${\hat{y}}_{\omega_{t}}$. Then, selecting $e_{3}$ as an ARR (Equation (17)), we can obtain residual-3 in state–space form, shown as Equation (18):(17)$$\;I_{p}\cdot {\dot{y}}_{\omega_{p}}+{\left(\frac{y_{\omega_{p}}}{KF\left(\frac{{\hat{y}}_{\omega_{t}}}{y_{\omega_{p}}}\right)}\right)}^{2}-y_{T}{}_{p}=0\;$$

(18)$$\;\{\begin{array}{l} \dot{x}=-\beta_{3}\left(x+I_{p}\cdot y_{\omega_{p}}\right)+{\left(\frac{y_{\omega_{p}}}{KF\left(\frac{{\hat{y}}_{\omega_{t}}}{y_{\omega_{p}}}\right)}\right)}^{2}-y_{T}{}_{p}\\ r_{3}=x+I_{p}\cdot y_{\omega_{p}} \end{array}\;$$

(4) Residual-4

Residual-4 is derived from $T_{4}$, which consists of seven equations: $\left\{e_{1},e_{2},e_{4},e_{5},e_{6},e_{7},e_{9}\right\}$.
(19)$$\;\{\begin{array}{l} e_{1}:T_{tp}={\left(\frac{\omega_{p}}{KF\left(SR\right)}\right)}^{2}\\ e_{2}:\;T_{tt}=TR\left(SR\right)\cdot T_{tp}\\ e_{4}:T_{t}=T_{tt}-{\dot{\omega }}_{t}\cdot I_{t}\\ e_{5}:SR=\frac{\omega_{t}}{\omega_{p}}\\ e_{6:}\;y_{\omega }{}_{p}=\omega_{p}\\ e_{7:}\;y_{\omega }{}_{t}=\omega_{t}\\ e_{9:}\;y_{T}{}_{t}=T_{t} \end{array}\;\;\;\;\;\;\;\;\;\;\;\;\;\;\;\;\;$$

Based on the same strategy of generating residual-1, we select $e_{4}$ as an ARR to obtain:(20)$$\;I_{t}\cdot {\dot{y}}_{\omega_{t}}+y_{T}{}_{t}-TR\left(SR\right)\cdot {\left(\frac{y_{\omega_{p}}}{KF\left(SR\right)}\right)}^{2}=0\;\;$$

Residual-4 can be designed by this ARR in state–space form, given by:(21)$$\;\{\begin{array}{l} \dot{x}=-\beta_{4}\left(x+I_{t}\cdot y_{\omega_{t}}\right)+y_{T}{}_{t}-TR\left(SR\right)\cdot {\left(\frac{y_{\omega_{p}}}{KF\left(SR\right)}\right)}^{2}\\ r_{4}=x+I_{t}\cdot y_{\omega_{t}} \end{array}\;$$

(5)  Residual-5

Residual-5 is derived from $T_{5}$ which consists of six equations: $\left\{e_{1},e_{3},e_{5},e_{6},e_{7},e_{8}\right\}$.
(22)$$\;\{\begin{array}{l} e_{1}:T_{tp}={\left(\frac{\omega_{p}}{KF\left(SR\right)}\right)}^{2}\\ e_{3}:T_{p}=T_{tp}+{\dot{\omega }}_{p}\cdot I_{p}\\ e_{5}:SR=\frac{\omega_{t}}{\omega_{p}}\\ e_{6:}\;y_{\omega }{}_{p}=\omega_{p}\\ e_{7:}\;y_{\omega }{}_{t}=\omega_{t}\\ e_{8:}\;y_{T}{}_{p}=T_{p} \end{array}\;\;\;\;\;\;\;\;\;\;\;\;\;\;\;\;\;$$

Referring to the same strategy of generating residual-1 and residual-4, we select $e_{3}$ as an ARR to obtain:(23)$$\;I_{p}\cdot {\dot{y}}_{\omega_{p}}+{\left(\frac{y_{\omega_{p}}}{KF\left(SR\right)}\right)}^{2}-y_{T}{}_{p}=0\;\;$$

Residual-5 can be designed by this ARR in state–space form, given by:(24)$$\;\{\begin{array}{l} \dot{x}=-\beta_{5}\left(x+I_{p}\cdot y_{\omega_{p}}\right)+{\left(\frac{y_{\omega_{p}}}{KF\left(SR\right)}\right)}^{2}-y_{T}{}_{p}\\ r_{5}=x+I_{p}\cdot y_{\omega_{p}} \end{array}\;$$

## 4. Design and Validation of the FDI System for the HTC

### 4.1. Establishment of the FDI System

In order to verify the correctness of the above FD and FI analysis by the SA-based methodology, this section will utilize a present torque converter model in the demos from MATLAB [45] and set up a fault detection and identification (FDI) system by the above five residuals, and then test them by numerical simulations in MATLAB Simulink. Figure 8 displays a block diagram of the FDI system, where four signals—two speed sensors ($y_{\omega }{}_{p},y_{\omega }{}_{t}$) and two torque sensors ($y_{T}{}_{p},y_{T}{}_{t}$)—from the system are inputted to the FDI system, as well as six injected faults ($f_{KF},f_{\omega_{p}\;},f_{\omega_{t}\;},f_{T_{p}\;},f_{T}{}_{t},f_{TR}$). Then, the proposed five residuals from the FDI system are evaluated in the residual observer, to judge whether there is a fault in the residual, and what kind of fault it is. 

Figure 9 displays the HTC characteristics with K-factor and torque ratio. Figure 10 shows the gear shifting and related velocity of the HTC system without faults, where the main parameters are set as follows:$\;I_{p}=0.012\;\mathrm{kg}\cdot m^{2}$, $I_{t}=0.01\;\mathrm{kg}\cdot m^{2}$, $\beta_{1}$
*=* 20, $\beta_{2}=80$,$\;\beta_{3}=50$, $\beta_{4}\;$= 10,$\;\beta_{5}=10$.

### 4.2. Validation of the FDI System for the HTC

#### 4.2.1. Fault Setting

To test the FDI system, we purposely inject six faults into the system, as shown in Figure 10a. Table 5 gives the designed fault type and duration of occurrence, where the fault type and time of occurrence are also assumed.

#### 4.2.2. Simulation and Discussion of the FDI System

After setting the faults in Table 5, we can obtain the signal output of the five residuals in the FDI system. Figure 11 shows the response of residual-1 to residual-5. Here, we set a fixed threshold as the standard to determine whether a fault exists. If the signal exceeds the threshold, a fault occurs; on the contrary, if the signal is below the threshold, there is no fault.

From Figure 11a, we know that residual-1 can detect faults $f_{\omega_{p}\;},f_{\omega_{t}\;},f_{T_{p}\;},f_{T}{}_{t},f_{TR}$, but not fault $f_{KF}$. In Figure 11b, residual-2 can detect faults $f_{KF},f_{\omega_{t}\;},f_{T_{p}\;},f_{T}{}_{t},f_{TR}$, but not fault $f_{\omega_{p}\;}$. In Figure 11c, residual-3 can detect faults $f_{KF},f_{\omega_{p}\;},f_{T_{p}\;},f_{T}{}_{t},f_{TR}$, but not fault $f_{\omega_{t}\;}$. In Figure 11d, residual-4 can detect faults $f_{KF},f_{\omega_{p}\;},f_{\omega_{t}\;},f_{T}{}_{t},f_{TR}$, but not fault $f_{T_{p}\;}.$ In Figure 11e, residual-5 can detect faults $f_{KF},f_{\omega_{p}\;},f_{\omega_{t}\;},f_{T_{p}\;},f_{TR}$, but not fault $f_{T}{}_{t}.$ In Figure 11f, residual-5 can also detect faults $f_{KF},f_{\omega_{p}\;},f_{\omega_{t}\;},f_{T_{p}\;},f_{T}{}_{t}$, but not fault $f_{TR}$.

Table 6 summarizes the five residuals’ detecting results, which are consistent with the theoretical analysis in Table 4 by the SA methodology. Thus, we may conclude that SA-based fault diagnosis of the HTC is applicable and feasible.

In Table 6, the symbol “●” indicates that the fault is detectable; “×” means that the fault is not detectable.

## 5. Experimental Validation

In this part, we utilized the experimental data from Pohl‘s research [8] for a quick and preliminary validation. In Pohl’s experimental study, he only tested a single HTC, instead of the HTC in an automatic transmission. Figure 12 shows the structure of the test rig in Pohl’s paper, where he installed four sensors to measure the input pump torque ($T_{p}$) and the pump speed ($\omega_{p}$), as well as the output turbine torque ($T_{t}$) and the turbine speed ($\omega_{t}$).

Note: There are three kinds of torque converter in Pohl’s study; we only used the data of the Honda CRV.

Because we only had the measurements of the four sensors ($y_{T_{p}}$, $y_{T_{t}}$,$\;y_{\omega_{p}}$,$\;y_{\omega_{t}}$), we executed an offline experimental validation; that is, we injected four sensor faults into the healthy signals, and then investigated the responses of the five residuals to judge whether they can detect the injected sensor faults.

Note: Here, we cannot study and observe the responses of the five residuals when the system faults ($f_{KF}$, $f_{TR}$) occur, because we have not established our own test stand, and we cannot inject the two faults into the HTC system. This work will be our next plan. The purpose of employing Pohl’s example here is to preliminarily examine our SA methodology and the related proposed FDI system.

Figure 13 shows the HTC characteristics of the Honda CRV in Pohl’s experiment [8]. Figure 14 gives the healthy signals of the four sensors, $y_{T_{p\;}}$, $y_{T_{t\;}}$,$\;y_{\omega_{p}}\;$,$\;y_{\omega_{t}}$.

Note: In Figure 13b, we use the American units, because we copied the data from Pohl’s paper [8]. We will transfer the torque to the international system of units in our subsequent residual testing.

Table 7 shows the injected sensor fault settings. Based on the FDI system in the simulation in Section 4.1, and after simulating the four sensor faults at the four different times, we obtained the responses of the five residuals, shown in Figure 15. Here, the parameters in the FDI system under the experimental environment are: ${I^{\prime }}_{p}=0.0926\;\mathrm{kg}\cdot m^{2},$
${I^{\prime }}_{t}=0.0267\;\mathrm{kg}\cdot m^{2}$, ${\beta^{\prime }}_{1}$ = 20, ${\beta^{\prime }}_{2}=1$, ${\beta^{\prime }}_{3}=10$, ${\beta^{\prime }}_{4}$ = 10, ${\beta^{\prime }}_{5}$ = 10.

From Figure 15a, we can observe that when fault $f_{\omega }{}_{p}$ occurs, only residual-2 cannot detect it. From Figure 15b, when fault $f_{\omega }{}_{t}$ occurs, only residual-3 cannot detect it. From Figure 15c, when fault $f_{T_{p}}$ occurs, only residual-4 cannot detect it. From Figure 15d, when fault $f_{T_{t}}$ occurs, only residual-5 cannot detect it. Table 8 shows a summary of the residual responses, which are consistent with the theoretical results in Table 4, as well as with the simulation results in Table 6. Thus, the experimental test again proved that the SA methodology is feasible in the fault diagnosis and FDI design for the HTC.

## 6. Conclusions

In this paper, we present a structural analysis (SA)-based fault detection and identification (FDI) scheme for the hydraulic torque converter (HTC). After executing the FMEA on the HTC, we obtained eight critical faults, and then we used two fault variables to represent them. To realize the maximal capability of FD and FI for the HTC, we employed the techniques of DM decomposition, FD, and FI in SA. Consequently, four available sensors were chosen to obtain maximal fault detectability and isolability: that is, all six faults are detectable, and five fault sets are isolable, leaving four faults that are uniquely isolable, and one group of sensor sets that is isolated from the other faults, but not from each other. The related results are shown in Figure 7 and Table 4. To realize the residual design, five MSO sets were obtained by SA; these sets generated five residuals. Then five robust residuals were designed based on the MSO sets and an ARR strategy. The proposed FDI scheme was realized in MATLAB Simulink, and the residuals were tested by injecting six faults in the HTC system. The results show that the five residuals can detect all the injected faults, which is consistent with the theoretical analysis; related results are clearly represented in Figure 11 and Table 6. Finally, we utilized experimental data and executed offline testing of SA-based fault diagnosis and FDI system design. The experimental results demonstrated consistency with the simulation, so that the effectiveness of the proposed approach is validated. Details are shown in Figure 15 and Table 8.

One noticeable contribution of the paper is that we proved that an SA-based FDI scheme is effective and efficient in performing fault diagnosis analysis and a FDI system design for the HTC. In the future, we plan to conduct an online experimental validation of the HTC, and to look for other mechatronic systems to be applied in the SA methodology.

## Figures and Tables

**Figure 1 sensors-18-04103-f001:**
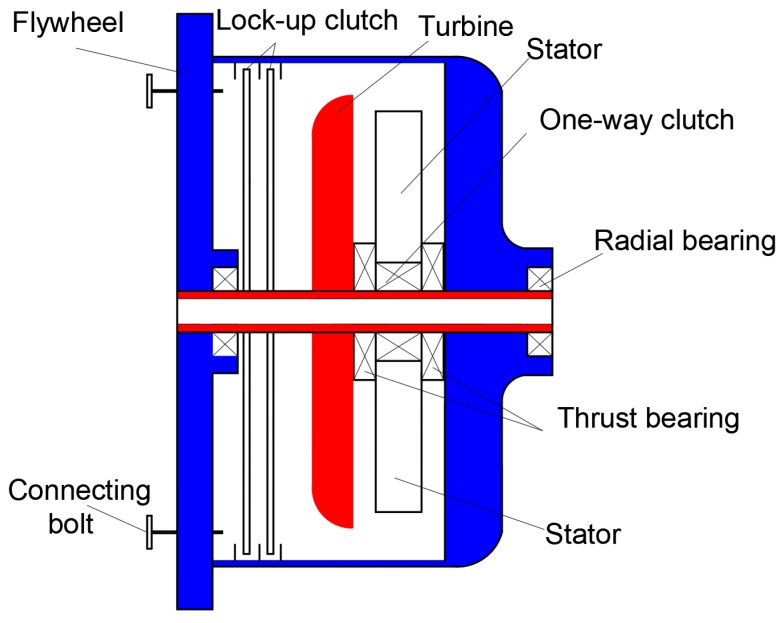
Structure of torque converter.

**Figure 2 sensors-18-04103-f002:**
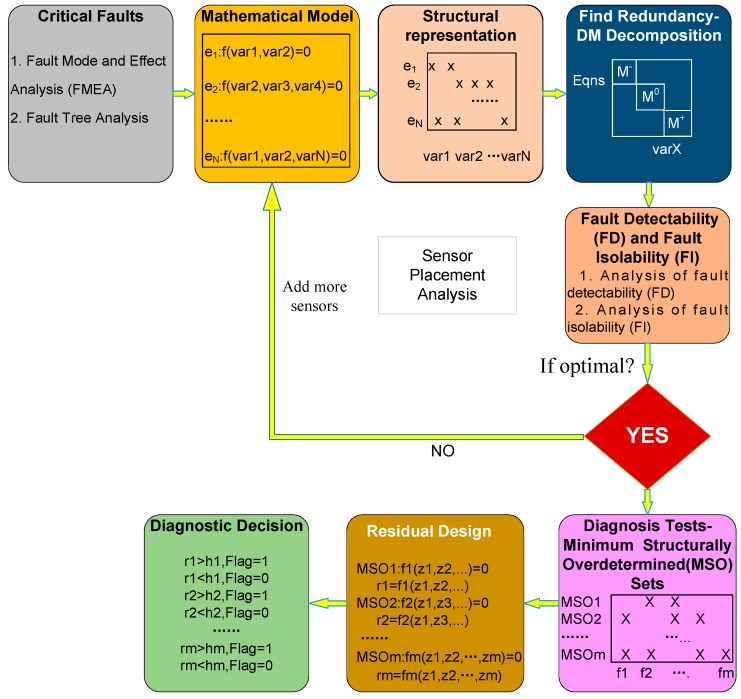
Main steps of structural analysis.

**Figure 3 sensors-18-04103-f003:**
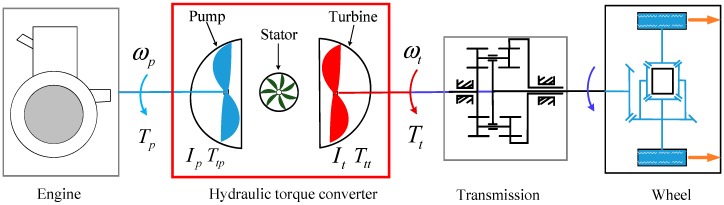
Typical structure of the hydraulic torque converter (HTC) in a vehicle. $\omega_{p}$ is the pump wheel angular velocity, $\omega_{t}$ is the turbine angular velocity, $T_{p}$ is the torque in the pump, $T_{t}$ is the turbine output torque, $T_{tp}$ is the torque transferred by the pump (impeller), $T_{tt}$ is the torque transferred by the turbine,$\;I_{p},\;I_{t}$ are the inertia of the pump and turbine, respectively.

**Figure 4 sensors-18-04103-f004:**
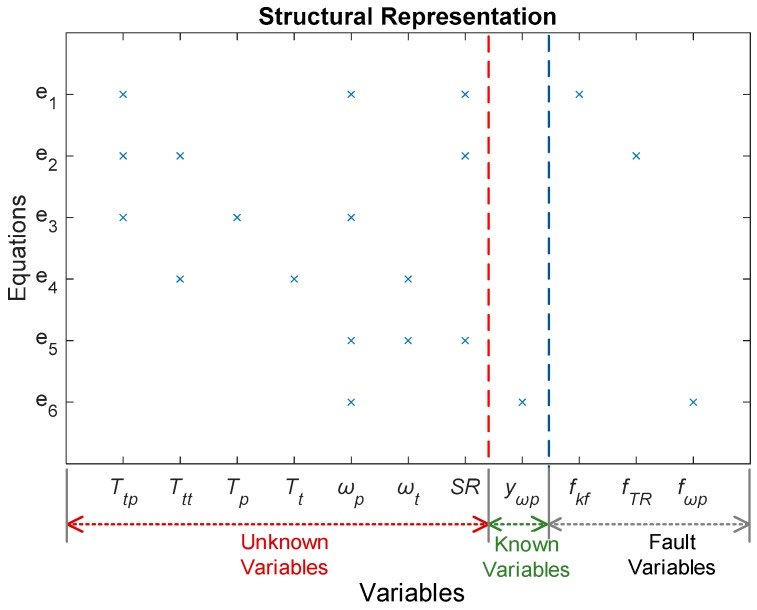
Structural representation of the HTC, with variables classified into three categories: unknown, known, and fault variables; “×” indicates that the equation *e_i_* (= 1–6) is associated with the corresponding variables.

**Figure 5 sensors-18-04103-f005:**
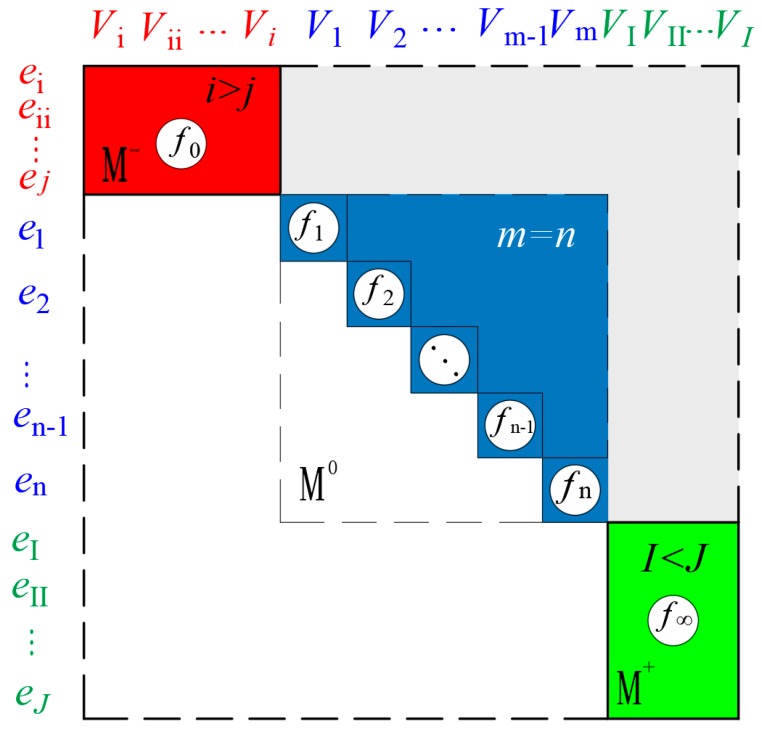
Diagram and principle of the Dulmage–Mendelsohn (DM) decomposition, where $e_{i}-e_{j}$,${\;e}_{1}-e_{n}$,${\;e}_{I}-e_{J}$ represent the equations, $V_{i}-V_{j}$,$\;V_{1}-V_{n}$,$\;V_{I}-V_{J}$ represent the system variables, and $f_{0}-f_{\infty }$ represent the fault variables.

**Figure 6 sensors-18-04103-f006:**
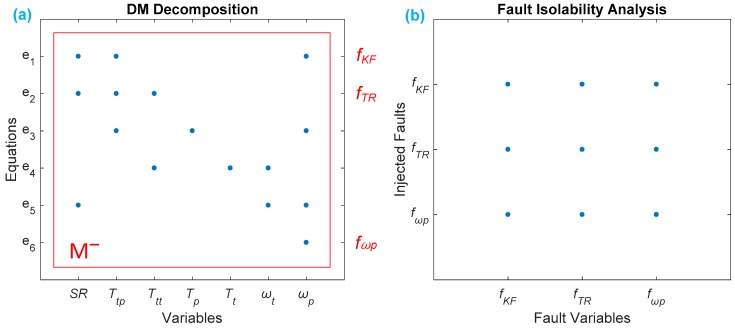
Fault detectability and isolability of the HTC: (**a**) DM decomposition for the HTC, where all the faults are located in the $M^{-}$ part, so all three faults are not detectable; (**b**) fault isolability matrix (FIM) of the HTC; “●” means the horizontal and longitudinal fault variables are correlated; thus, all three faults are not isolatable.

**Figure 7 sensors-18-04103-f007:**
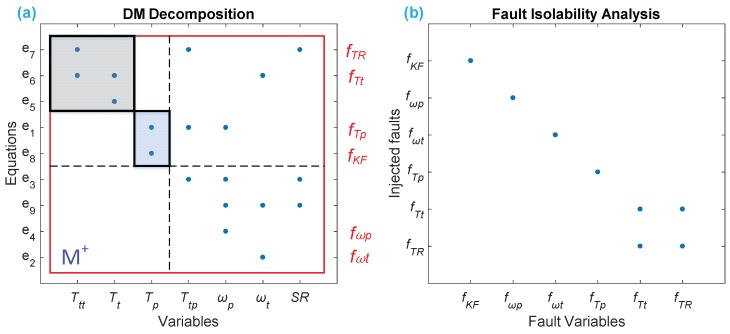
Result of FD and FIM of placing four sensors on $\omega_{p}$,$\;\omega_{t}$, $T_{p}$, and $T_{t}$: (**a**) DM decomposition, where all six faults are detectable, because they all lie in $M^{+}$; (**b**) FIM of the HTC, where four faults are uniquely isolable, leaving two faults isolated from the others, but none isolable from each other.

**Figure 8 sensors-18-04103-f008:**
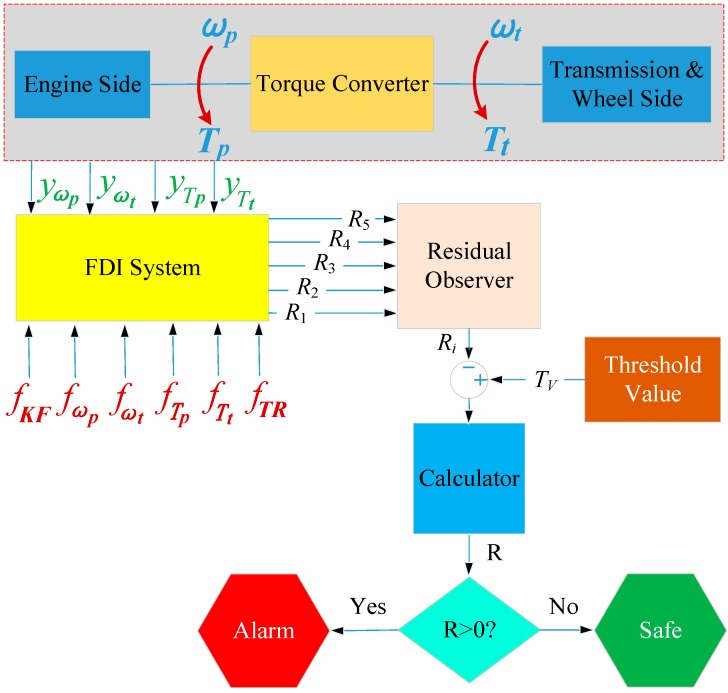
Diagram of the fault detection and identification (FDI) system.

**Figure 9 sensors-18-04103-f009:**
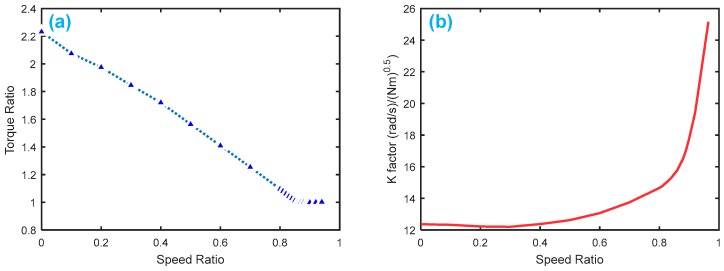
Torque converter characteristics: (**a**) torque ratio by speed ratio; (**b**) K-factor by speed ratio.

**Figure 10 sensors-18-04103-f010:**
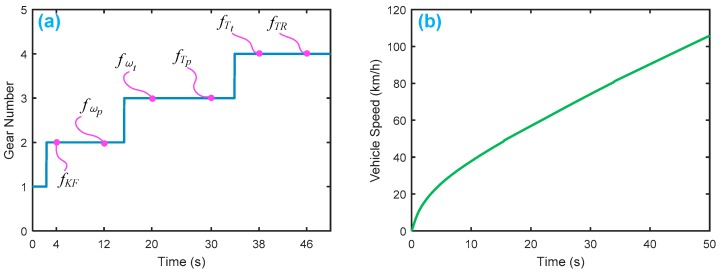
Normal driving simulation of a vehicle with an HTC in a four-speed transmission: (**a**) gear shifting in a driving cycle; (**b**) vehicle speed in a driving cycle. Here, we only show gradual acceleration without braking.

**Figure 11 sensors-18-04103-f011:**
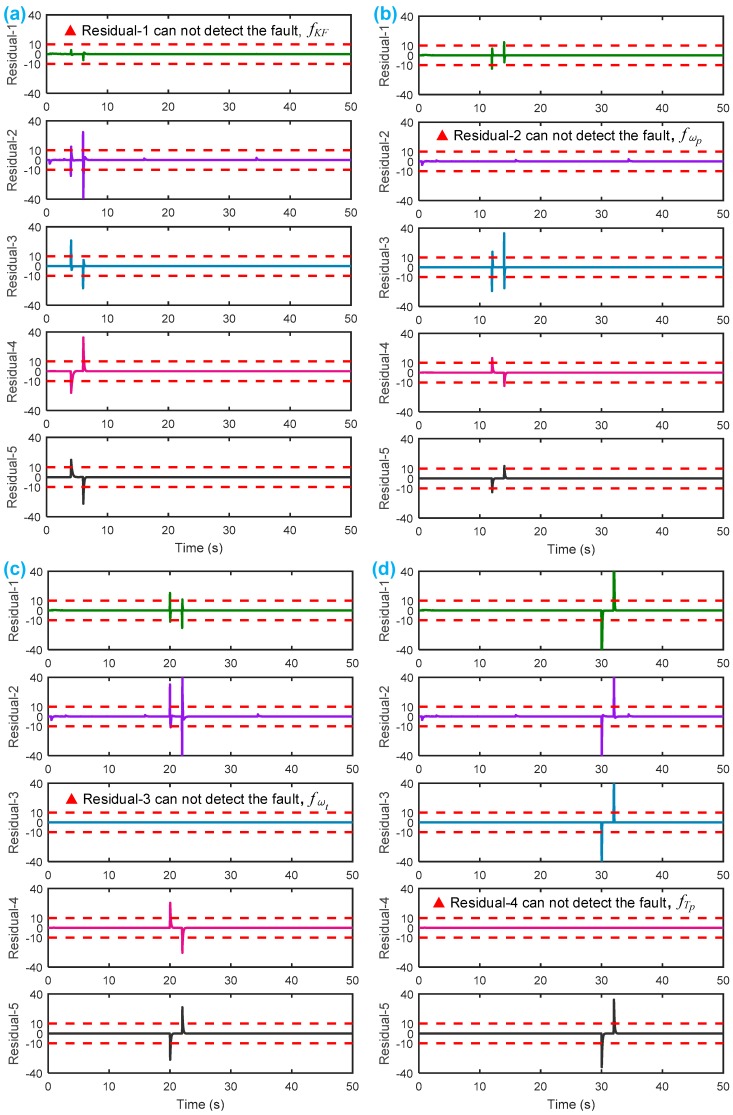
Responses of the five residuals: (**a**) when fault $f_{KF}$ happens; (**b**) when fault $f_{\omega_{p}}$ happens; (**c**) when fault $f_{\omega_{t}}$ happens; (**d**) when fault $f_{T_{p}}$ happens; (**e**) when fault $f_{T_{t}}$ happens; (**f**) when fault $f_{TR}$ happens.

**Figure 12 sensors-18-04103-f012:**
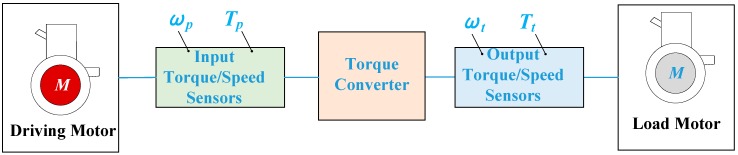
Principle diagram of the test stand in Pohl’s study [8].

**Figure 13 sensors-18-04103-f013:**
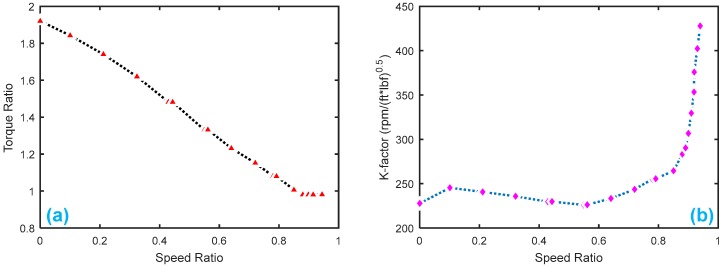
Torque converter characteristics of the HTC in a Honda CRV from Pohl’s paper: (**a**) torque ratio by speed ratio; (**b**) K-factor by speed ratio.

**Figure 14 sensors-18-04103-f014:**
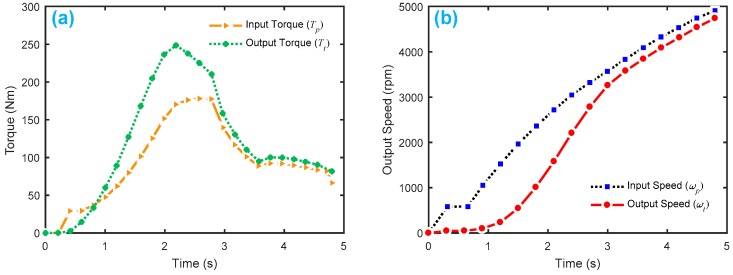
Measurement of four sensors in Pohl’s experiment: (**a**) input torque ($T_{p}$) and output torque ($T_{t}$); (**b**) input speed ($\omega_{p}$) and output speed ($\omega_{t}$).

**Figure 15 sensors-18-04103-f015:**
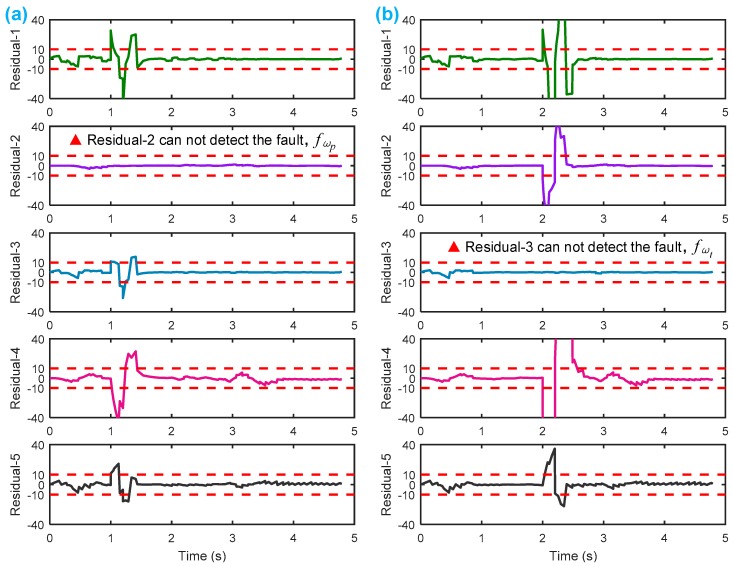
Responses of the five residuals for the HTC in a Honda CRV from Pohl’s experiment: (**a**) when fault $f_{\omega_{p}}$ happens; (**b**) when fault $f_{\omega_{t}}$ happens; (**c**) when fault $f_{Tp}$ happens; (**d**) when fault $f_{Tt}$ happens.

**Table 1 sensors-18-04103-t001:** Fault mode and effect analysis (FMEA) results of hydraulic torque converter. S, severity; O, occurrence; D, detection; RPN, risk priority number; ATF, automatic transmission fluid.

Fault Code	Part	Fault Mode (Code)	Fault Causes	Fault Effects	*S*	*O*	*D*	RPN
*F* _101_	Blade of pump wheel	Distortion	Excessive load, fatigue, impurities/debris in ATF oil	Decrease/loss of power, larger vibration, abnormal noise	6	2	6	72
		Fracture/broken			9	2	5	90
*F* _102_	Radial bearing at pump wheel	Deformation/fracture	Excessive load, fatigue, lack of lubrication, excessive temperature	Decreased pump speed and torque, vibration, abnormal noise	5	2	6	60
		Burn			8	1	5	40
*F* _103_	Shaft neck at pump wheel	Surface abrasion	High friction with copper sleeve, lack of lubrication	Oil leakage, decreased torque output, vibration, abnormal noise	5	3	5	75
		Axial fracture	Overload, fatigue, improper installation	No power, vibration, abnormal noise, no gear shifting	*6*	*2*	*4*	48
*F* _104_	Blade of turbine wheel	Distortion	Excessive load, fatigue, impurities/debris in ATF oil	Decrease/loss of power, more fuel, larger vibration, abnormal noise	7	2	6	84
		Fracture/broken			9	3	5	135
*F* _105_	Spline of turbine wheel	Wear	Overload, fatigue	Decrease/loss of power, frustration in driving the vehicle, no gear shifting	6	2	6	72
		Broken			*8*	*3*	*5*	120
*F* _106_	Axle sleeve of turbine wheel	Scratch	Overload, wear, fatigue, lack of lubrication	Excessive swing in torque converter	3	2	7	42
		Wear			*4*	*2*	*7*	56
*F* _107_	Guide ring in stator	Block	ATF oil deteriorated, overload, iron filings gathered	Slow start of vehicle, decrease/loss of power, larger vibration	7	2	6	84
		Broken			*8*	*3*	*5*	120
*F* _108_	One-way clutch in stator	Deformation	Overload, over-speed, fatigue, lack of lubrication	Decrease/loss of power in turbine speed and torque, bigger vibration, abnormal noise	4	3	6	72
		Fracture			*8*	*2*	*5*	80
*F* _109_	Thrust bearing in stator	Deformation	Fatigue, overload, over-speed, lack of lubrication	Decreased power to turbine wheel, more fuel, larger vibration, abnormal noise	5	2	6	60
		Fracture			*8*	*1*	*5*	40
*F* _110_	Connection bolt between pump wheel and flywheel	Loose	Improper installation, fatigue, overload	Abnormal noise, jitter, unstable/loss of output speed torque	6	2	6	72
		Fall off			*9*	*2*	*5*	90
*F* _111_	Seal ring	Deformation	Flywheel or drive plate deformation, sleeve strain in outer bearing surface, ATF oil deterioration	Torque converter leakage, insufficient torque converter output power	5	2	6	60
		Wear/corrosion			*8*	*2*	*5*	80
*F* _112_	Lock-up clutch	Slipping	Overload, fatigue, wear, locking friction plate fell off	Powerless at high speed, too high temperature, shuddering, low efficiency	6	2	6	72
*F* _113_	Friction plate in lock-up clutch	Abrasion	Overload, fatigue, excessive temperature	Decreased torque in turbine, more fuel, powerless at medium and high speed	5	3	6	90
		Fracture			6	2	5	60
*F* _114_	Solenoid valve in lock-up clutch	Leakage	No power/short circuit, insufficient oil pressure, link fracture	Lock-up clutch connection failure, lock-up clutch separation stuck	5	3	6	90
		No reaction			*8*	*3*	*5*	120

**Table 2 sensors-18-04103-t002:** Fault variables and their types, and the relationships between variables and critical faults in the HTC.

Fault Symbol	Related Faults	Type
$f_{KF}$	Pump group Pump blade fracture, damaged seals/oil leakage, separated connection between pump wheel and flywheel, stuck lock-up clutch separation	Gain
$f_{TR}$	Turbine group Turbine blade fracture, spline broken in the turbine wheel, damaged guild ring in stator, lock-up clutch connection failure	Gain

**Table 3 sensors-18-04103-t003:** Results of the sensor placement for the HTC.

Group#	Sensor No.	FT.No.	Det.FT.No.	UnDet.FT.No.	Iso.FT. Set.No.	Uni.Iso.FT.No.	UnDet.Fault. List	Iso.Fault Sets List	Unique Iso.Fault List	Sensor List
1	2	4	0	4	0	0	$f_{KF},f_{TR},f_{\omega p},f_{TP}$			$T_{P}$
2	2	4	0	4	0	0	$f_{KF},f_{TR},f_{\omega p},f_{Tt}$			$T_{t}$
3	2	4	0	4	0	0	$f_{KF},f_{TR},f_{\omega p},f_{\omega t}$			$\omega_{t}$
4	3	5	5	0	1	0		$f_{KF},f_{TR},f_{\omega p},f_{TP}$		$T_{P},T_{t}$
5	3	5	4	1	1	0	$f_{TR}$	$f_{KF},f_{\omega p},f_{Tp},f_{\omega t}$		$T_{P},\omega_{t}$
6	3	5	5	0	1	0		$f_{KF},f_{TR},f_{\omega p},f_{Tt}$		$T_{t},\omega_{t}$
7	4	6	6	0	5	4		$f_{TR},f_{Tt}$	$f_{\omega p},f_{\omega t},f_{Tp},f_{KF}$	$T_{P},T_{t},\omega_{t}$

**Table 4 sensors-18-04103-t004:** Minimal structurally overdetermined (MSO) sets of the HTC.

	$\;f_{KF}\;$	$\;f_{\omega }{}_{p}\;$	$\;f_{\omega }{}_{t}\;$	$\;f_{T}{}_{p}\;$	$\;f_{T_{t}}\;$	$\;f_{TR}\;$	Equations	
$T_{1}$	×	●	●	●	●	●	MSO1	$e_{2},e_{3},e_{4},e_{5},e_{6},e_{7},e_{8},e_{9}$
$T_{2}$	●	×	●	●	●	●	MSO2	$e_{1},e_{2},e_{3},e_{4},e_{5},e_{7},e_{8},e_{9}$
$T_{3}$	●	●	×	●	●	●	MSO3	$e_{1},e_{2},e_{3},e_{4},e_{5},e_{6},e_{8},e_{9}$
$T_{4}$	●	●	●	×	●	●	MSO4	$e_{1},e_{2},e_{4},e_{5},e_{6},e_{7},e_{9}$
$T_{5}$	●	●	●	●	×	×	MSO5	$e_{1},e_{3},e_{5},e_{6},e_{7},e_{8}$

**Table 5 sensors-18-04103-t005:** Injected faults setting in the HTC.

Fault	Type	Signal	Time Span
$f_{KF}$	Gain (1.5)	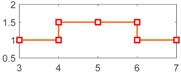	4–6 s
$f_{\omega_{p}}$	Bias (−150)	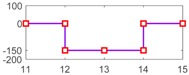	12–14 s
$f_{\omega_{t}}$	Bias (+200)	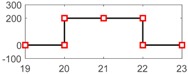	20–22 s
$f_{T_{p}}$	Gain (2)	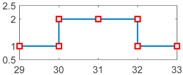	30–32 s
$f_{T_{t}}$	Gain (0.5)	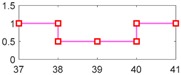	38–40 s
$f_{TR}$	Gain (0)	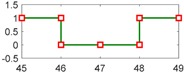	46–48 s

**Table 6 sensors-18-04103-t006:** Summary of detecting the results of the five residuals by the proposed FDI system.

-	$\;f_{KF}\;$	$\;f_{\omega }{}_{p}\;$	$\;f_{\omega }{}_{t}\;$	$\;f_{T}{}_{p}\;$	$\;f_{T_{t}}\;$	$\;f_{TR}\;$
$R_{1}$	×	●	●	●	●	●
$R_{2}$	●	×	●	●	●	●
$R_{3}$	●	●	×	●	●	●
$R_{4}$	●	●	●	×	●	●
$R_{5}$	●	●	●	●	×	×

**Table 7 sensors-18-04103-t007:** Injected sensor fault settings of the HTC in a Honda CRV from Pohl’s experiment.

Fault	Type	Signal	Time Span
$f_{\omega_{p}}$	Bias (+500)	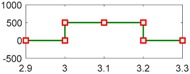	1–1.2 s
$f_{\omega_{t}}$	Bias (−1000)	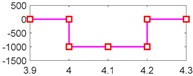	2–2.2 s
$f_{Tp}$	Gain (0)	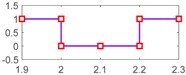	3–3.2 s
$f_{Tt}$	Gain (10)	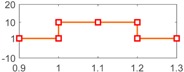	4–4.2 s

**Table 8 sensors-18-04103-t008:** Summary of the testing results of the five residuals.

	$\;f_{\omega }{}_{p}\;$	$\;f_{\omega }{}_{t}\;$	$\;f_{T}{}_{p}\;$	$\;f_{T_{t}}\;$
$R_{1}$	●	●	●	●
$R_{2}$	×	●	●	●
$R_{3}$	●	×	●	●
$R_{4}$	●	●	×	●
$R_{5}$	●	●	●	×

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
