# Peer review of "Design and Evaluation of a Structural Analysis-Based Fault Detection and Identification Scheme for a Hydraulic Torque Converter"

_sensors, 2018, doi:10.3390/s18124103_

Round 1
Reviewer 1 Report
This paper deals with Hydraulic torque converter (HTC) fault diagnosis, by combining the technique of fault mode and effect analysis (FMEA) which is the well-known bottom-up strategy to explore potential fault patterns and the principal of Analytical Redundancy which allows residual generation for fault detection and isolation. The severity of each fault (Severity, S), the occurrence frequency of the fault (Occurrence, O), and the detection of the fault (Detection, D) are evaluated and represented by numbers (1 to 10), and then risk priority number (RPN) by S×O×D is used to determine the risk level of each fault.
The combination of FMEA and AR is not common in the literature because the FMEA requires extensive expert knowledge and AR requires extensive physical knowledge, and both are often not available at the same time, but in this case study (HTC) These two types of knowledge are available which gives relevance to the proposed approach.
The paper is well written and easy to understand, and the application support is topical and will interest readers especially the expert knowledge given in Table 1. The experimental results are well illustrated and show the effectiveness of the method proposed in this paper.
However the review of the literature must be improved by paper review, as for example "A survey of fault diagnosis and fault-tolerant techniques; Part I: IEEE Transactions on Industrial Electronics 62 (6) (2015) 3757-3767
Another approach of structural analysis for fault diagnosis, based on bond graph methodology, must be cited in this work as it allows thanks to its causal and structural properties to browse a physical model like a directed graph, and to highlight therefore the relations of cause-effect or to confirm the cause-and-effect relationships derived from expert knowledge. For example, 'LFT bond graph model-based robust fault detection and isolation'. Bond graph modeling of engineering systems. PP. 105-133. 2011.
Reviewer 2 Report
The paper presents an application SA-based FDI scheme to HTC system, in which it covers the issues with detectability and isolability. These issues are hot subject in the field and received great attention over years.
However, the performances and feasibility have not been evaluated by an experiments that are consistent with the FDI system developed, making it difficult to justify the quality of the work. Furthermore, the contribution to knowledge is hard to be identified. These two aspect must be paid high attention in improving the manuscripts.
In addition, the following concerns are also need to be considered:
Are model based FDI for suspension system[18], the hydraulic braking system[19], the steer-by-wire systems[20,21], and the 51 electrical steering system[22], etc. based SA analysis? If so how they are different from current work?
Any references or particular methods for the risk priority numbers used?
How the FMEA table help to determine the eight critical faults. These eight faults described cause nearly full losses of HTC function or complete failure. It makes no sense to implement fault detection, rather other detecting incipient faults such as medium wear, small leakage, ATF oil deterioration can be meaningful in that detecting this fault can prevent the failures and ensure safety and efficient operation.
3.1 may not be an independent subsection, only some of the key steps/issues such as sensor placement in association detectability, isolability as well as estimation of severity of a fault may need to be revisited after have the mathematical models (3.2)
Similarly 3.4 describing Dulmage-Mendelsohn Decomposition, rather it can be introduced coherently in 3.5 but it should give a rigorous explanations on the graphs and explain how M+,M-and M0 are produced based on the model and variables of HTC system
Explain why needs to Find MSO Sets, and is there any other ways to do so?
Justify the rationality of the fault types of ‘gain’ and ‘biased’ induced to the HTC system in association with engineering knowledge.
The two validation systems (simulation and experiment from third party) all uses speed from turbine, which does not agree with the FDI system developed in that wheel speed is used.
A though proofreading is required to improve English expressions for better clearness.
Round 2
Reviewer 2 Report
The revisions are significant and acceptable.
Please consider to clarify the ‘two variables’ more as below,
‘By fault mode and effect analysis (FMEA), eight critical faults are identified in the HTC, which are further associated to two main groups or categories: pump and turbine ones respectively for clearness in discussion’
if this is what the meaning by the authors, please use such as pump group or turbine group where are suitable in the entire content
Please replace ‘detected’ with ‘detectable’ where are appreciated.
Figure 6(a), not Figure 6a, and so on
